**communications** engineering

# Industrial-scale prediction of cement clinker phases using machine learning
Sheikh Junaid Fayaz[1], Néstor Montiel-Bohórquez[2], Shashank Bishnoi[1], Matteo Romano [2],
Manuele Gatti [2] & N. M. Anoop Krishnan [1] ✉

Cement production exceeds 4.1 billion tonnes annually, emitting 2.4 billion tonnes of $CO_2$ annually, necessitating improved process control. Traditional models, limited to steady-state conditions, lack predictive accuracy for clinker mineralogical phases. Here, using a comprehensive two-year industrial dataset, we develop machine learning models that outperform conventional Bogue equations with mean absolute percentage errors of 1.24%, 6.77%, and 2.53% for alite, belite, and ferrite prediction respectively, compared to 7.79%, 22.68%, and 24.54% for Bogue calculations. Our models remain robust under varying operations and are evaluated for uncertainty and rare-event scenarios. Through post hoc explainable algorithms, we interpret the hierarchical relationships between clinker oxides and phase formation, providing insights into the functioning of an otherwise black-box model. The framework can potentially enable real-time optimization of cement production, thereby providing a route toward reducing material waste and ensuring quality while reducing the associated emissions under real-world conditions.

Global cement production reached more than 4.1 billion tons/year in 2023, more than doubling from 2005 levels[1,2], with projections indicating a further increase by 2050[2]. Moreover, each tonne of cement releases ~0.66 tonnes of $CO_2$[3], contributing to more than 8% of global carbon emissions. This increasing cement demand[4], along with high emissions, necessitates enhanced production efficiency while maintaining stringent quality standards. The performance of cement, particularly its 28-day compressive strength, is primarily governed by the clinker's mineralogical phases–alite, belite, aluminate, and ferrite. Alite drives early strength development, belite contributes to long-term strength[5], and ferrite influences color and provides minor early-age strength[6]. However, while alite-rich cements enhance early strength, alite content greater than 65%[7] can lead to increased heat of hydration and increased $CO_2$ emissions due to higher limestone requirements in the raw feed. Conversely, belite-rich cements exhibit improved long-term durability but may delay early strength gain due to lower reactivity. Therefore, controlling the relative composition of these phases is critical in determining the cement quality[8].

Quality assessment of clinker mineralogy traditionally relies on X-ray diffraction (XRD), performed either online with 15-30 minute delays or offline with up to 4-hour measurement cycles[9]. These delays result in considerable material waste when out-of-specification clinker is produced. Real-time prediction of clinker phases would not only eliminate this waste but also allow engineers to proactively adjust process parameters, ensuring the desired clinker composition is met before production, rather than reacting to delayed XRD results.

Several studies have applied first-principles methods (FPM) to model various aspects of cement manufacturing, including calciner dynamics[10], $NO_x$ formation via computational fluid dynamics (CFD)[11], ab-initio modeling of the electronic structure of $C_3S$[12], and density functional theory (DFT) based calculations of enthalpy and lattice parameters for clinker phases[13]. However, first-principles approaches for predicting clinker mineralogy remain limited and lack exploratory depth. For instance, Mastorakos et al.[14] modeled clinker formation using a 1D CFD-based dynamic kiln simulation without validation against plant data. Instead, the authors emphasize the model's qualitative correctness, showing that predicted clinker phases align with expected industrial ranges based on experience while omitting quantitative error metrics based on plant data.

While physics-based modeling has successfully addressed various aspects of cement production, including calciner operations[15,16], waste heat recovery[17,18], alternative fuel assessment[19], and $CO_2$ capture[20], accurate prediction of clinker mineralogy remains challenging. Hökfors et al.[21] proposed a raw-meal-based model for clinker phases; however, it neglects the influence of process parameters and is validated against only a single plant data point, leaving its capability for capturing complex plant-wide dynamics unexamined. A comprehensive physics-based approach would require detailed knowledge of process interactions, intermediate reaction mechanisms, and kinetic rate constants–factors difficult to determine in an operational setting. Furthermore, the high computational cost of physics-based models limits their feasibility for real-time phase estimation.

[1]Indian Institute of Technology Delhi, New Delhi, India. [2]Politecnico di Milano, Department of Energy, Milan, Italy. ✉e-mail: krishnan@iitd.ac.in

In light of the challenges with physics-based models, Moses et al.[22] proposed a regression model for predicting alite composition. However, it was developed solely on synthetic data–generated from literature models[14,23–25]–lacking real plant variability and uncertainties. Alternatively, recent advances in machine learning (ML)[26] demonstrate promising capabilities in predicting cement and concrete properties[26–29]. While these ML approaches have been validated on limited laboratory-scale data[27–29], recent study[30] demonstrated plant-level prediction of $C_3S$, $C_2S$, and $C_4AF$–idealized compounds derived from Bogue calculations. However, these purely theoretical compounds omit real-phase complexities and impurities, making them easier to model. In contrast, alite, belite, and ferrite are the actual phases in clinker, influenced by impurities and process variations, making their prediction more challenging and industrially relevant. Nonetheless, the study highlights ML's growing potential in the cement industry. Given the limitations of FPM, physics-based approaches, and empirical models in capturing clinker phase complexities, a fundamental question arises: can ML, by leveraging industrial-scale data, overcome these gaps and surpass traditional modeling in accuracy and adaptability—ultimately enabling real-time process optimization?

To address this challenge, we leverage a comprehensive two-year industrial dataset from an operating cement plant to develop predictive models for clinker mineralogical phases. Our framework achieves at least 88% lower mean absolute error compared to previously reported models[31] evaluated on real plant data. We rigorously evaluate the developed models on variable industrial conditions at a scale not explored in the literature thus far. The models consistently outperform traditional Bogue equations, offering a practical pathway toward automated process control. Through post-hoc explainability methods, we elucidate the quantitative relationships between clinker oxides and phase formation dynamics. Finally, we also develop plant-specific equations that considerably exceed conventional Bogue calculations in prediction accuracy, providing a quick assessment tool for plant-scale operations.

## Results

### Data for ML model development

A comprehensive two-year dataset from an operational cement plant provided the foundation for developing ML models in this study (details in Data collection, Methods. Data quality assessment revealed notable variability in clinker phase compositions. An exemplar variation in alite content for the entire duration is shown in Fig. 1b, along with the respective distribution (Fig. 1c, d). The two-year alite measurements exhibited a broad distribution (45–70 wt.%) with distinct temporal patterns. Similar variations were observed in belite (5–25 wt.%) and ferrite (11–17 wt.%) compositions (Fig. 1e–g). The complete distributions of all the features in DB1 and DB2 are included in Figs. S1–S4, Supplementary A.

To ensure model robustness, A 3-tier preprocessing protocol was implemented to address missing entries, duplicate values, and physically inconsistent measurements, and data outliers removal resulting in a curated dataset of 8654 clinker compositions. In the outlier removal values falling outside the 0.01-99.99 percentile range were classified as outliers and excluded from analysis, as illustrated by the shaded regions in Fig. 1b. This filtering criterion was uniformly applied across all 59 input features and 3 output variables. Additionally, the dataset's multi-scale temporal structure necessitated synchronization, as process parameters were recorded at high frequency (1-minute intervals), whereas material compositions had lower sampling rates. A systematic temporal alignment protocol was implemented, accounting for residence times across different production stages to establish meaningful correlations between process conditions and clinker composition. This rigorous preprocessing approach ensured a high-quality, complete dataset for subsequent analysis. The final dataset was split into training (80%) and test (20%) sets, maintaining the temporal distribution of phase compositions (Fig. 1e–g, shown in green and yellow respectively). Details on data collection, preprocessing, and synchronization are provided in the Methods section.

The preprocessed dataset exhibit distribution for all major phases, with alite centered at $\mu = 60.3$ wt.% ($\sigma = 3.2$), belite at $\mu = 14.9$ wt.% ($\sigma = 3.2$), and ferrite at $\mu = 14.3$ wt.% ($\sigma = 0.8$) (see Fig 1e–g). These distributions confirm that the data in the plant is indeed unbiased and is coming from a single distribution, confirming that operating conditions did not undergo any major change in the two-year period considered. Thus, this data could be reliably used for developing predictive models while capturing the inherent variability in industrial operations.

### Performance of ML models for clinker phases

Input feature selection critically influences model performance and practical utility in industrial settings. While comprehensive plant data theoretically offers maximum information content, a parsimonious model utilizing strategically selected input parameters may achieve comparable accuracy. The design of data-driven models were governed by two key aspects: accurate prediction of clinker mineralogy, and potential to implement model predictive. While there are limitations no the input parameters while considering the first aspect, the second requires the input to be directly controllable during the plant operation so as to control the clinker mineralogy and hence quality.

We categorized potential input combinations into two distinct classes: predictive control features (process parameters and raw materials) enabling real-time process optimization and post-production analysis features incorporating clinker oxide measurements. The latter, while potentially offering superior accuracy due to enriched chemical and process information, remains unsuitable for online process control due to inherent measurement delays and availability of the features post-production. To systematically evaluate these trade-offs, we constructed 15 distinct feature sets (Fig. 1a), encompassing various combinations of kiln feed characteristics, process parameters, hot meal properties, and clinker oxide compositions. This comprehensive approach enabled the identification of minimal yet sufficient feature sets for accurate phase prediction while maintaining practical applicability for process control.

We systematically evaluated nine machine learning architectures for clinker phase prediction using the complete feature set of 59 features from KF, HM, PP, and CO. Figure 2 compares the performance of linear models (linear regression, lasso, ridge, elastic net) against non-linear approaches (random forest, XGBoost, support vector regression (SVR), Gaussian process regression (GPR), neural network (NN)[32–36]) across three primary clinker phases (see Methods for details). Table 1 shows the performance of all the models for alite, belite and ferrite. Each model underwent rigorous cross-validation to ensure optimal hyperparameter selection and prevent overfitting. Note that, in the study separate models have been used to predict each clinker phase rather than multi-output models. This allows independent hyper-parameter tuning of each model, optimizing its performance for the specific target phase.

For all 3 phase predictions (Fig. 2a–c), linear models consistently demonstrated higher mean absolute percentage errors (MAPE), clustering near the radar plot periphery. This performance deficit persisted despite the inclusion of comprehensive process parameters, suggesting inherently non-linear relationships between input features and phase composition. This observation challenges the industry-standard Bogue equation, which assumes linear relationships between clinker oxides and phases. Non-parametric models–particularly NN, GPR, and SVR–achieved superior prediction accuracy across all three phases, with MAPEs of 1.24%, 6.77%, and 2.53% for alite, belite, and ferrite, respectively (see Fig. 2a–c).

### Benchmarking against Bogue equation

To benchmark against industry standards, we compared our models against the plant-specific Bogue equation (Fig. 2d–i), obtained directly from the domain experts in the specific plant. A two-month period, comprising January and February in 2020, was considered for this evaluation, which was kept unseen by the model and was excluded from the training data. Thus, this period is equivalent to an unseen operating period

**Fig. 1 | Dataset characteristics and temporal variability in clinker phases. a** Schematic representation of a cement plant showing key measurement locations: kiln feed (KF), process parameters (PP), hot meal (HM), and clinker oxides (CO). The Venn diagram illustrates the combinations of input features used for model development. **b** Two-year temporal evolution of alite content showing plant variability (black dots) with 0.01-99.99 percentile bounds (green shading). **c, d** Frequency distribution of alite content in the complete dataset and zoomed view highlighting the normal distribution. **e–g** Time series and distribution analysis of major clinker phases (alite, belite, ferrite) showing data partitioning into training (yellow) and test (green) sets. Each subplot includes temporal evolution (left) and frequency distribution (right) with mean (μ) and standard deviation (σ) values. The full dataset is shown in red, the training set in yellow, and the test set in green.

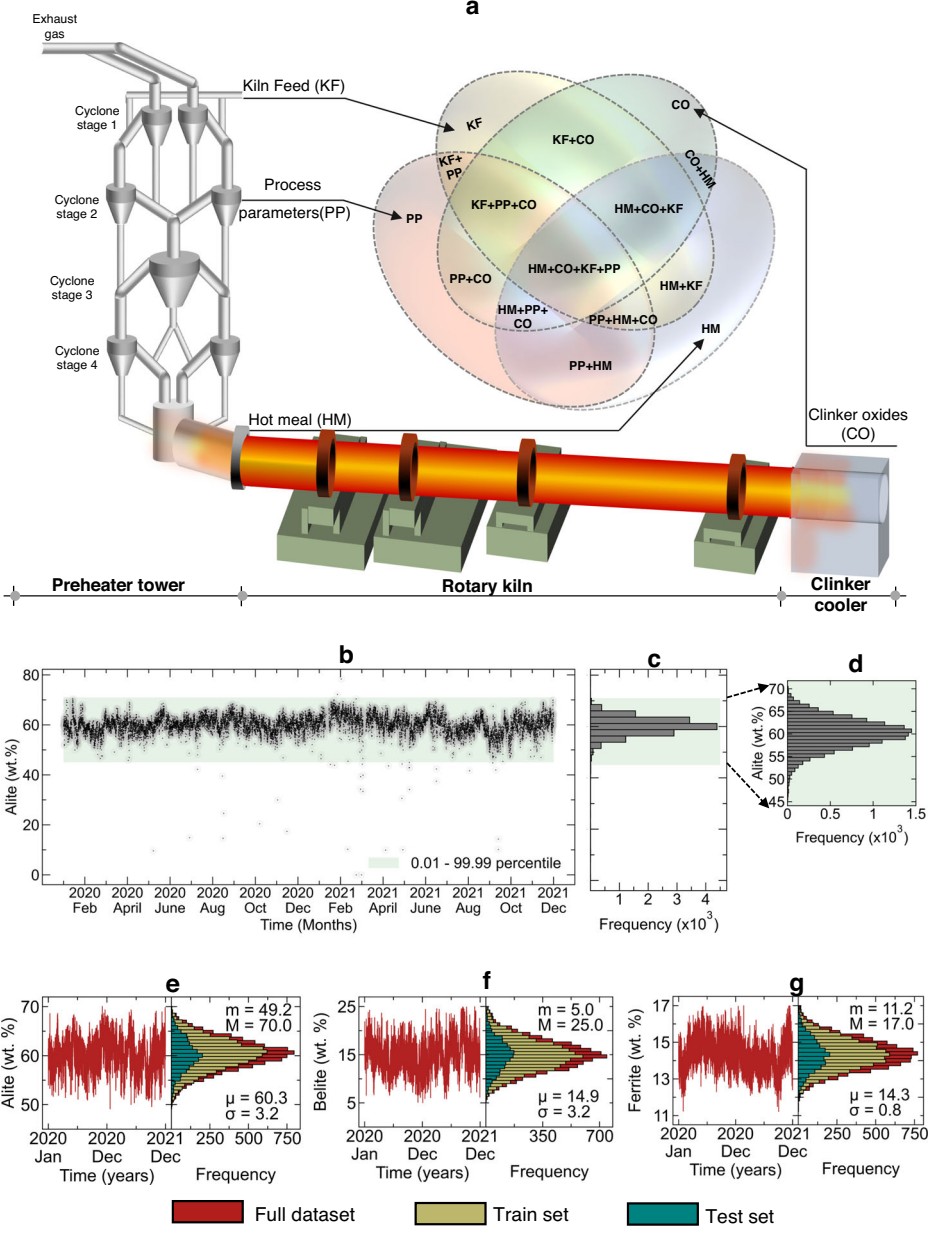

of the plant and serves as a true test of the ML model. The best-performing architectures corresponding to each of the phases as identified from Fig. 2a–c were used. ML model robustness was evaluated through 20 independent training iterations using different random seeds. Figure 2 presents the mean predictions obtained from these 20 trained models. The gray bands enveloping the mean predictions in Fig. 2e, g, i represent the uncertainty in model predictions arising from variations across the 20 training runs. Performance of the best-performing models, along with error bars representing uncertainty, is shown in Fig. S10, Supplementary C. The best-performing architectures, namely, NN for alite, GPR for belite, SVR for ferrite, demonstrated remarkable improvements over Bogue predictions. Parity plots (Fig. 2d, f, h) reveal tighter clustering around the ideal prediction line, while temporal predictions over a two-month test period (Fig. 2e, g, i) show that the models capture complex compositional dynamics with high fidelity. Error distributions of the models (inset histograms) confirm substantially reduced prediction variance compared to Bogue calculations. Further, statistical analysis validated the superior performance of ML models over traditional Bogue

calculations. ML predictions demonstrated symmetric error distributions (Fig. 2d), contrasting with Bogue's systematic biases. ML predictions, especially NN, SVR and GPR, showed exceptional consistency across training and test sets, confirming robust generalization across the entire compositional range.

Notably, the ML models accurately tracked rapid compositional fluctuations while maintaining ± 3σ prediction confidence intervals (grey regions). Notably, some of the spikes in the alite, with compositional variations of ~15 wt.% in a day, were captured accurately by the ML model. In contrast, the Bogue equation exhibited systematic biases: overestimating alite content (MAPE: 7.79% vs 1.24%), underestimating belite composition (MAPE: 22.68% vs 6.77%), and severely misrepresenting ferrite concentrations (MAPE: 24.54% vs 2.53%). These results underscore the limitations of linear approximations in capturing complex clinker formation dynamics. More importantly, the results conclusively demonstrate the superior ability of ML models to predict the clinker compositions accurately despite the huge fluctuations, making them a promising tool for online model predictive control.

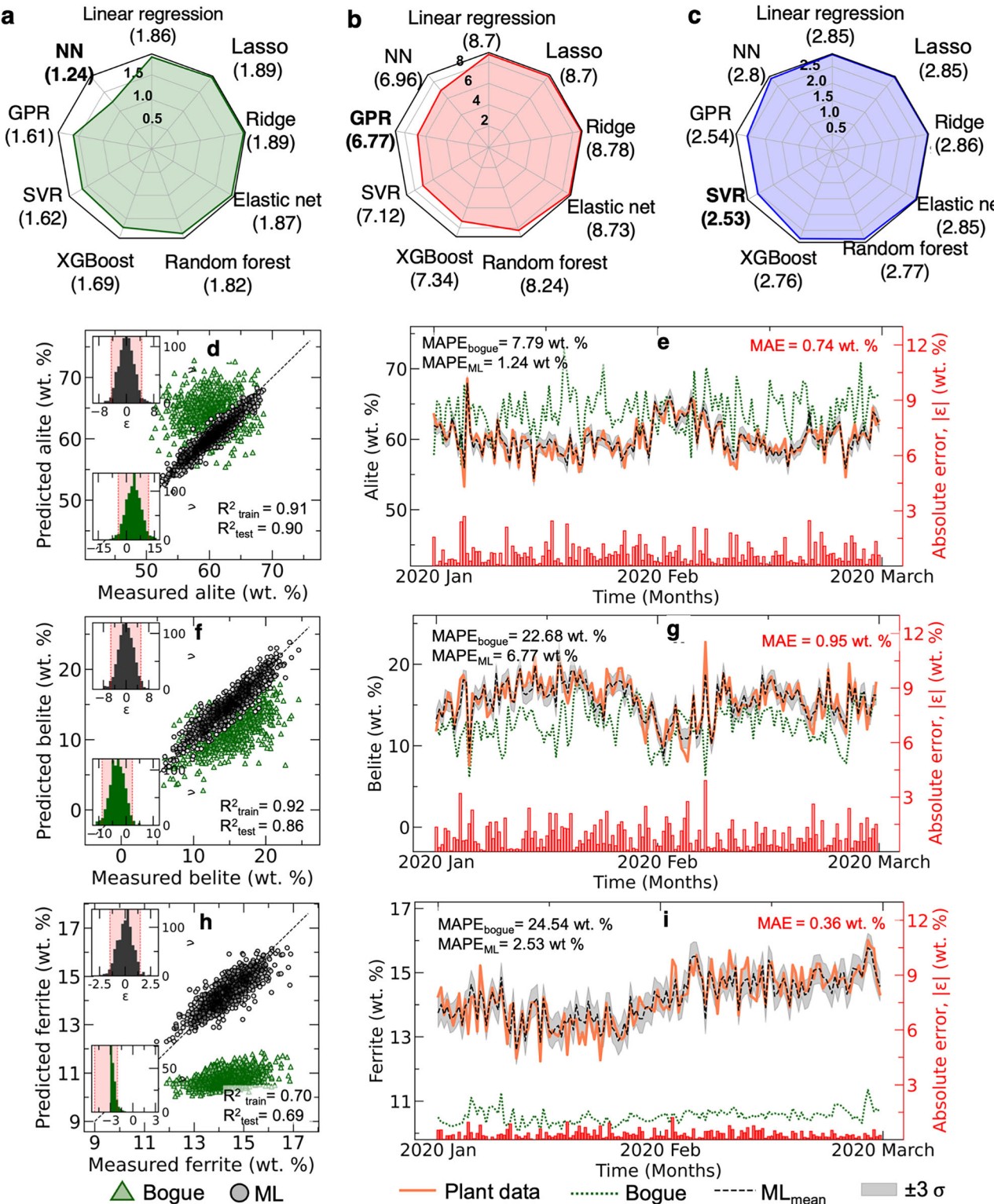

**Fig. 2 | Performance comparison of machine learning architectures for clinker phase prediction.** Mean Absolute Percentage Error (MAPE) across nine ML models for predicting (**a**) alite, (**b**) belite, and (**c**) ferrite compositions using complete feature sets (KF, PP, HM, CO), respectively. Values in parentheses indicate test-set MAPE. The best-performing models are shown in bold. Quantitative performance metrics ($R^2$ and MAPE) for the best-performing model against traditional Bogue calculations represented as parity plot and temporal evolution, respectively, for (**d**, **e**), alite, (**f**, **g**), belite, and (**h**, **i**), ferrite with inset histograms showing error distributions ($\epsilon$ = predicted - actual) for ML models (top) and Bogue calculations (bottom). Red-shaded regions in histograms represent 95% confidence intervals ($\pm 2\sigma$), with x-axis limits set at 99.9% confidence ($\pm 4\sigma$). The temporal evolution of predictions is over a two-month test period showing plant data (red), ML predictions (black dashed), and Bogue calculations (green dotted). Grey bands represent model uncertainty ($\pm 3\sigma$), while red bars (right axis) indicate absolute prediction errors. All error metrics are reported in weight percentage (wt.%).

**Table 1 | Comparing the performance of ML models on the test set using MAE, $R^2$ and MAPE**

| Models | Alite | | | Belite | | | Ferrite | | |
|---|---|---|---|---|---|---|---|---|---|
| | MAE(wt.%) | $R^2$ | MAPE (%) | MAE(wt.%) | $R^2$ | MAPE (%) | MAE(wt.%) | $R^2$ | MAPE (%) |
| Linear regression | 1.11 | 0.79 | 1.86 | 1.23 | 0.76 | 8.70 | 0.40 | 0.63 | 2.85 |
| Lasso | 1.11 | 0.79 | 1.86 | 1.23 | 0.76 | 8.70 | 0.40 | 0.63 | 2.85 |
| Ridge | 1.13 | 0.78 | 1.89 | 1.24 | 0.75 | 8.78 | 0.41 | 0.63 | 2.86 |
| Elastic net | 1.12 | 0.78 | 1.87 | 1.23 | 0.76 | 8.73 | 0.40 | 0.63 | 2.85 |
| Random forest | 1.10 | 0.79 | 1.82 | 1.15 | 0.78 | 8.24 | 0.39 | 0.64 | 2.77 |
| XGBoost | 1.01 | 0.82 | 1.69 | 1.03 | 0.83 | 7.34 | 0.39 | 0.65 | 2.76 |
| SVR | *0.96* | *0.84* | *1.62* | *0.99* | *0.84* | *7.12* | **0.36** | **0.69** | **2.53** |
| GPR | <u>0.96</u> | <u>0.84</u> | <u>1.61</u> | **0.96** | **0.86** | **6.77** | <u>0.36</u> | 0.70 | <u>2.54</u> |
| NN | **0.75** | **0.90** | **1.24** | <u>0.98</u> | <u>0.84</u> | <u>6.96</u> | 0.40 | 0.65 | *2.80* |

The scores for the best, second best, and third best models are shown using bold, underlined, and italics, respectively.

- **Plant specific equations:** To establish a fair comparison between Bogue equation, we developed plant-specific linear regression models (clinker equations) using identical input parameters as Bogue equations (Fig. 3b–g).Clinker equations were developed for four cases (see Supplementary D), and their performance in predicting clinker phases is shown in Fig. 3(a). The MAPE across all four cases shows minimal variation.These tailored equations demonstrated marked improvements over standard Bogue calculations, particularly for alite ($R^2_{test} = 0.51$ vs $R^2_{test} = 0.23$) and belite ($R^2_{test} = 0.33$ vs $R^2_{test} = 0.26$) predictions. The temporal evolution plots (Fig. 3c, e, g) reveal substantially reduced mean absolute errors: 1.77 wt.%, 2.12 wt.%, and 0.49 wt.% for alite, belite, and ferrite respectively. This analysis conclusively demonstrates that even simplified plant-specific models offer substantially more reliable quality control metrics than traditional Bogue calculations. Thus, instead of relying on traditional Bogue equations, having simplified plant-specific equations obtained purely in a data-driven fashion can serve as a better performance indictor to be used for quality control. Details of the equations and their derivations are presented in Supplementary D.

- **Input feature pruning:** To identify the best combination of input features that balance between clinker prediction accuracy and predictive control, we systematically evaluated model performance across 15 distinct feature combinations through a radial visualization (Fig. 4a). The full-feature models (PP+KF+HM+CO) achieved optimal accuracy with MAPEs of 1.24%, 2.53%, and 6.77% for alite, ferrite, and belite, respectively. However, these post-production predictions, while accurate, offer limited utility for real-time process control. Specifically, any model with CO as input features inhibit predictive control as CO measurements are obtained only post-production. Thus, the remaining set of 7 models, with input features excluding CO were analyzed.

We observed that models without CO exhibit reasonable performance, albeit slightly poorer than those with CO as input feature. However, all the models performed better than the Bogue equation. Notably, even reduced-feature ML models using only process parameters outperformed Bogue calculations: alite (MAPE: 3.14% vs 7.79%), ferrite (3.48% vs 24.54%), and belite (7.18% vs 22.68%). This suggests that a predictive control implemented purely based on PP can outperform those based Bogue, not to mention that Bogue requires the clinker oxide compositions which can be obtained only post-production. The gradual degradation in prediction accuracy with feature reduction is clearly visualized in the radial plot, providing process engineers with a quantitative framework for feature selection based on specific accuracy requirements.

- **Sparse ML models:** While maximal-information ML models leveraging all 59 input features– including process PP, KF, HM, and CO –demonstrate exceptional predictive accuracy for clinker phase composition, their practical implementation could be challenging. Many process parameters exhibit strong correlations (see Fig. S5, Supplementary A), and obtaining a complete dataset with all 59 features in an industrial setting may

not always be feasible. To address this, we propose parsimonious ML models which can predict alite, belite, and ferrite phases with MAPE of 2.91%, 11.41%, and 3.22%, respectively (Fig. 4b–d). The models rely on a minimal number of inputs, utilizing only 10 independently controllable PP (marked with * in Table S1, Supplementary A) along with KF composition. The selected 10 PP are independent and exhibit low mutual correlation (see Fig. S5, Supplementary A), while the remaining 24 process parameters are dependent variables, determined as a consequence of plant operations governed by these 10 controllable inputs. Despite using a reduced input set, the parsimonious ML model maintains high predictive accuracy, making it both practical and interpretable. While slightly less accurate than maximal-information models, it substantially outperforms Bogue and the developed clinker equations, demonstrating ML's potential in data-limited industrial settings.

### Model limitations: rare event scenarios

While the ML models demonstrate remarkable accuracy in predicting clinker phase compositions, it is crucial to assess their limitations, particularly their performance under extreme, rare plant operating conditions with very low occurrence frequency. The models were trained on a preprocessed dataset, free from anomalies and extreme values, as outliers beyond the 0.01-99.99 percentile range were removed. As shown in Fig. 5a–c, the number of phase composition data points outside this filtering window is minimal (26 for alite, 104 for belite, and 58 for ferrite). Figure 5d, e further illustrates that the probability of these rare compositions is below 1%, with the plant operating within the normal range for over 99% of the time. However, when tested on these rare events, the models exhibit notable deviations, leading to a high spike in prediction errors, as shown in Fig. 5g, h. This highlights a fundamental limitation of data-driven models–their inability to extrapolate beyond the training range. Nonetheless, given that the plant operates within the normal range for the vast majority of the time, the developed ML models remain highly effective for quality control and process optimization in routine operations.

### Interpreting the ML models

To bridge the gap between model performance and domain expertise, we employed SHAP (SHapley Additive exPlanations) analysis to quantify feature contributions and their directional impact on clinker phase predictions. We focused our analysis on clinker oxide (CO) features, excluding process parameters (PP) due to their complex interdependencies, to enable direct comparison with established mineralogical understanding.

The hierarchical influence of clinker oxides emerges distinctly in Fig. 6a–c. For alite prediction, CaO and $SiO_2$ demonstrate dominant influence, with mean absolute SHAP values of 1.6 and 0.8 wt.%, respectively. The beeswarm visualization (Fig. 6d) reveals that increased CaO content (63–67 wt.%) positively correlates with alite formation, while elevated $SiO_2$

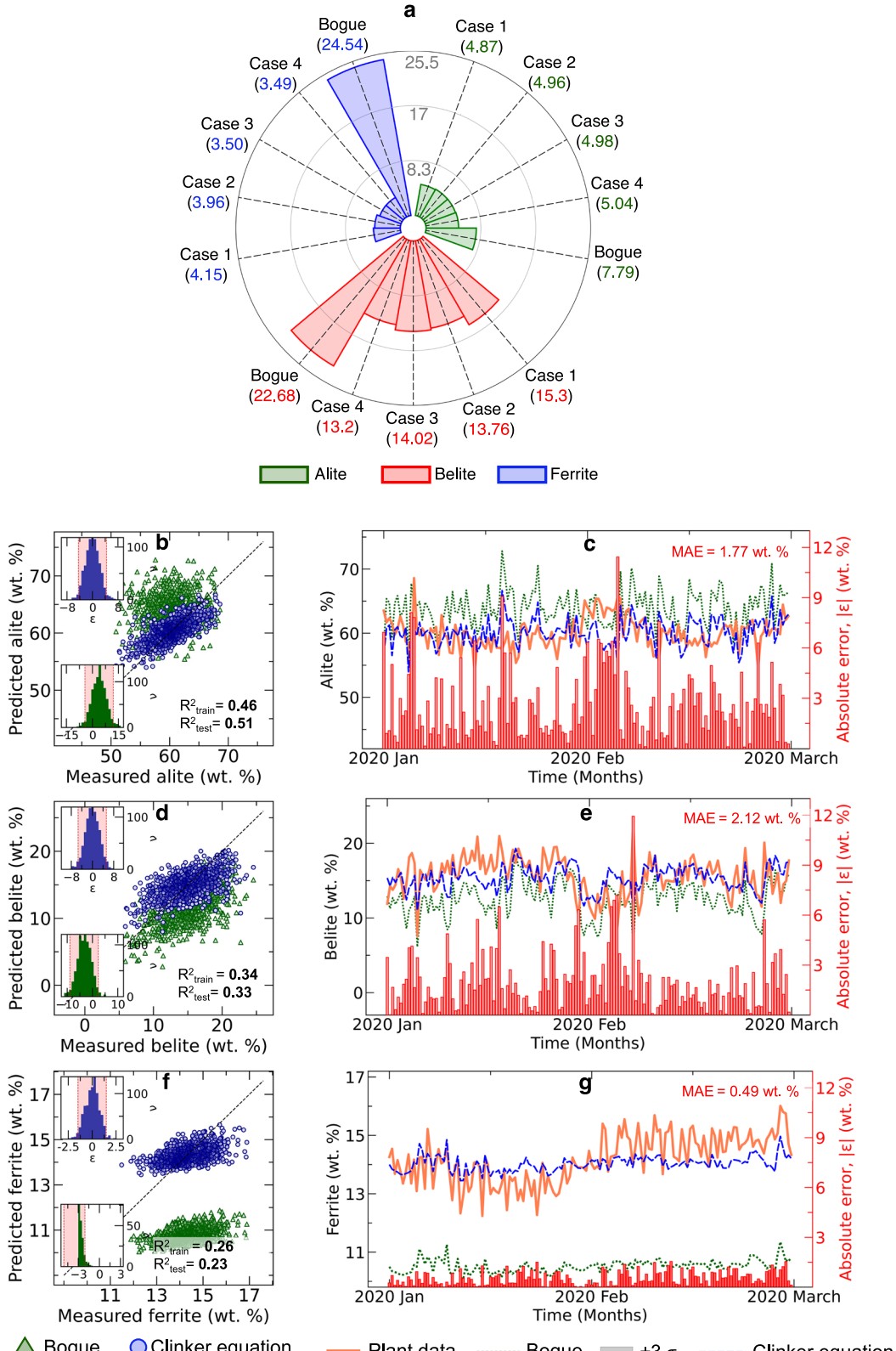

**Fig. 3 | Plant-specific clinker equations. a** Comparison of clinker equations formulated in 4 cases with Bogue's equation for predicting clinker phases. The radial direction shows the MAPE (%.) values. **b–g** Performance evaluation of plant-specific clinker equations against standard Bogue calculations. Parity and temporal plots comparing predicted versus measured compositions for (**b, c**) alite, (**d, e**) belite, and (**f, g**) ferrite, respectively. Inset histograms show error distributions for clinker equations (top) and Bogue calculations (bottom). The temporal evolution of predictions is over a two-month test period showing plant data (red), clinker equation predictions (blue dashed), and Bogue calculations (green dotted). Grey bands represent model uncertainty ( ± 3σ), while red bars (right axis) indicate absolute prediction errors. Training ($R^2_{train}$) and test ($R^2_{test}$) set performance metrics demonstrate superior accuracy of plant-specific equations over traditional Bogue calculations. All compositions and errors are reported in weight percentage (wt.%).

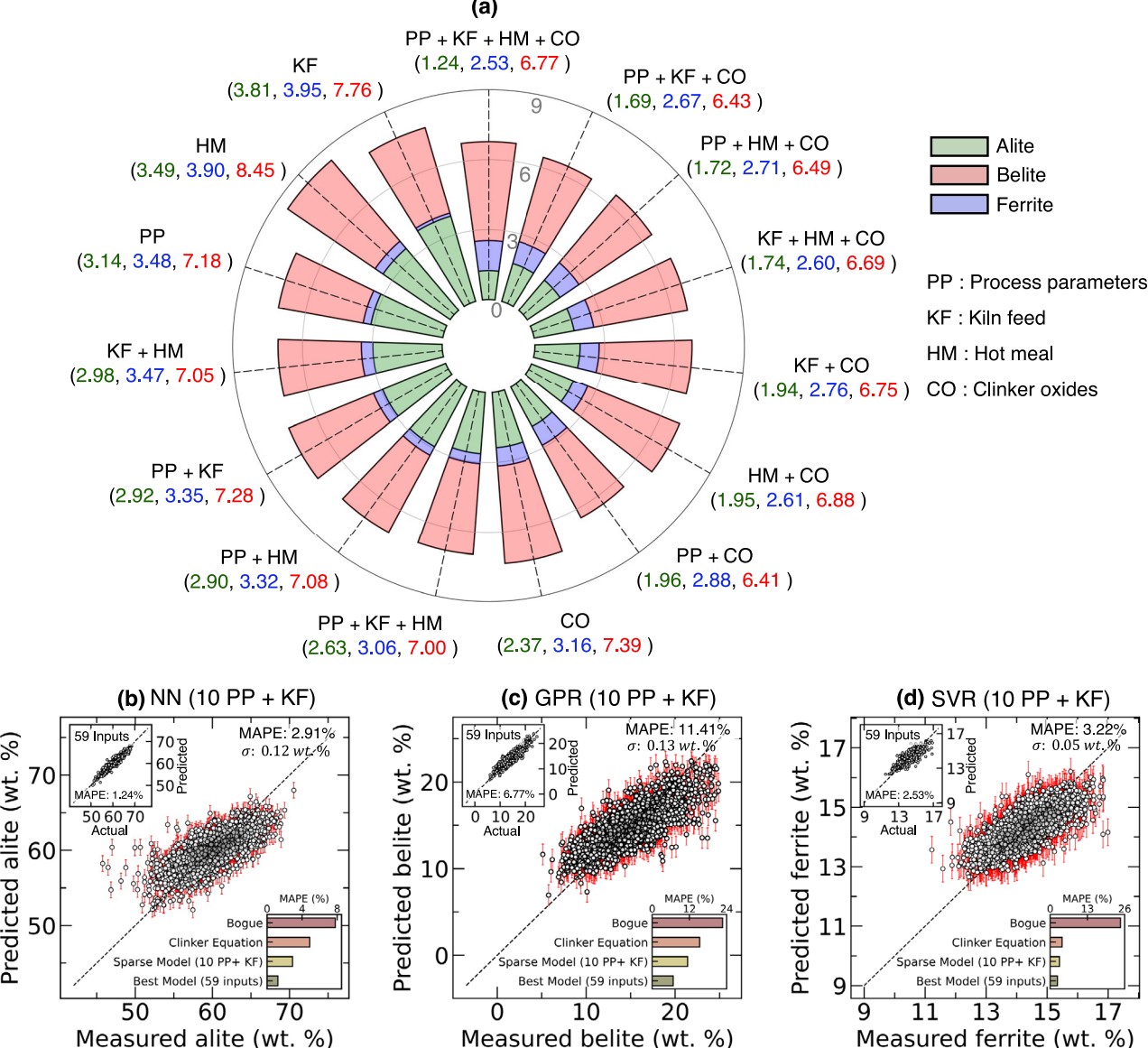

**Fig. 4 | Sparse ML model development via input feature pruning for clinker phase. a** MAPE of optimal machine learning models (Neural Network for alite, Gaussian Process Regression for belite, Support Vector Regression for ferrite) across 15 combinations of input features: process parameters (PP), kiln feed (KF), hot meal (HM), and clinker oxides (CO). Values in parentheses represent MAPE (%) for alite (green), ferrite (blue), and belite (red) predictions. **b** Alite prediction using NN, (**c**) Belite prediction using GPR, and (**d**) Ferrite prediction using SVR, utilizing 10 controllable PP and KF compositions. The top-right inset shows each phase's best-performing model--NN for alite, GPR for belite, and SVR for ferrite--when trained with all 59 input features. Black circles represent the mean model-prediction, while red error bars indicate model uncertainty ( $\pm 3\sigma$ ). The bottom-left inset presents a comparative analysis of MAPE for the sparse model against the best-performing model, as well as the Bogue and clinker equations developed in this study.

levels (20-22 wt.%) exhibit negative correlation, aligning with classical clinker chemistry principles.

Belite predictions (Fig. 6b, e) show similar oxide dependencies, though with distinct quantitative relationships. CaO maintains primary influence (mean |SHAP values| = 1.64 wt.%), followed by $SiO_2$ and $Na_2O$. The ferrite phase demonstrates unique sensitivity to $Fe_2O_3$ content (mean |SHAP values| = 0.38 wt.%), with alkali oxides ($Na_2O$, $K_2O$) exhibiting secondary influence (Fig. 6c, f), as $Fe_2O_3$ primarily governs and represents the ferrite phase formation.

It is also worth noting that both the importance of the features and the directionality of their influence (positive vs negative) are congruent with the plant-specific and the Bogue equations. Thus, the SHAP-derived relationships corroborate that the relationship learned by the ML models is congruent with the established domain knowledge while

providing quantitative insights into feature interactions. The analysis confirms that our ML models capture fundamental physicochemical relationships governing clinker phase formation, enhancing their credibility for industrial deployment.

## Discussion

Given the limitations of existing studies (Table 2), this work establishes a new paradigm in cement manufacturing by achieving remarkably high accuracy in predicting clinker mineralogical phases at an industrial scale, rigorously validated on large-scale plant data. Our computational framework substantially outperforms conventional Bogue equations across all metrics while maintaining predictive accuracy even with minimal input parameters, facilitating both instantaneous monitoring and retrospective analysis. Moreover, the present work can provide on-the-fly predictions of

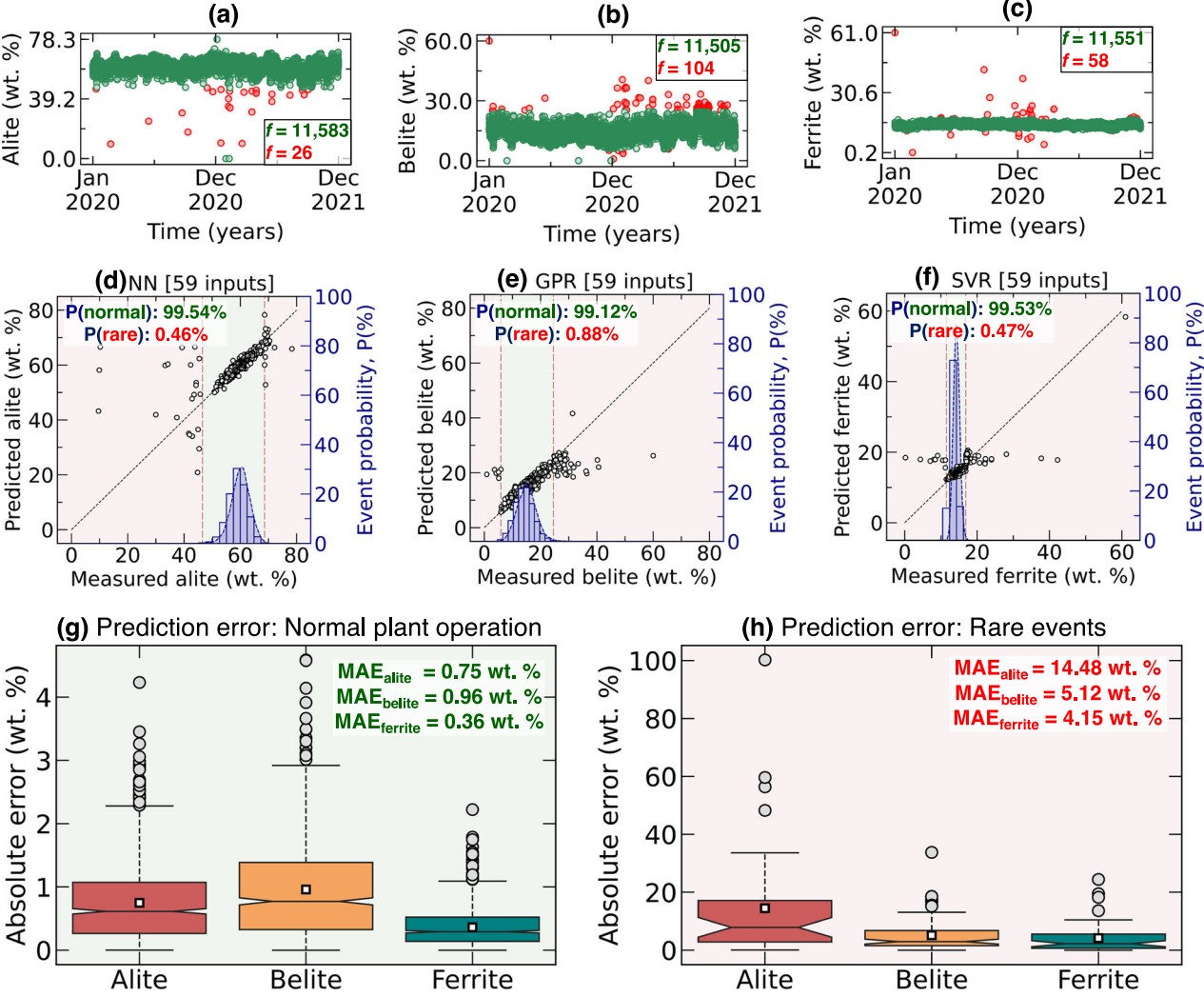

**Fig. 5 | Extreme event model evaluation.** Distribution of (**a**) alite, (**b**) belite, and (**c**) ferrite over two years of plant operation. Data retained after preprocessing is shown in green, while data removed during the three-tier outlier removal process is shown in red. The red-marked events correspond to rare plant operations with very low occurrence frequency, denoted as *f* for each phase. **d**–**f** illustrate the performance of the best-performing models across normal operation ranges (green background) and rare operation conditions (red background). The total probability of normal and rare operations is also indicated for each phase. The prediction errors for (**g**) normal operation and (**h**) extreme operation are shown using box plots of absolute errors.

the clinker compositions, while Bogue equation is a post-mortem analysis based on the clinker oxides.

The results presented provide key insights for industrial cement production in three critical dimensions. First, ML models can provide accurate estimates for clinker phase prediction, a feat that has not been possible thus far based on large-scale operational plant data with huge fluctuations. Second, ML models utilizing solely operational parameters and raw material compositions can provide pre-production estimations, enabling the development of robust quality assurance protocols and potential for online process control. Third, facility-specific equations derived from operational measurements markedly exceed standard calculations, offering pragmatic intermediate solutions during digital transformation initiatives.

While this study focuses on predictive modeling of clinker mineralogy, deploying these models in industrial settings is crucial for realizing their broader implications in advancing sustainability goals. However, plant demonstration presents several challenges, outlined below as future research directions.

- **Latency due to Model fine-tuning**: As plant operations can evolve beyond the range of the training data used in the models, continuous fine-tuning with new operational data would be necessary to ensure

efficient adaptation with evolving plant operations. Various parallelization techniques can be evaluated on GPUs to accelerate fine-tuning time. Also, to minimize cloud-based latency, models will be deployed on local industrial PCs or edge devices near kiln control systems.

- **Integrating models with existing Control Systems** requires: (a) developing an API to enable real-time communication between the ML model and control systems, ensuring structured model outputs compatible with logic controllers managing kiln processes; (b) implementing real-time dashboards for operators to monitor predictions, receive alerts, and take corrective actions; and (c) establishing fail-safe mechanisms with fallback logic that reverts to traditional control methods if the ML model encounters issues, ensuring uninterrupted plant operations.

- **Handling Sensor Noise and Data Delays**: Real-time outlier detection and filtering pipelines will be implemented to manage noisy sensor inputs. For temporary sensor failures, interpolation, last-known values, or suitable imputation methods will be used. Time synchronization, based on process delays, will ensure proper alignment of sensor data collected from different process stages.

**Fig. 6 | Feature attribution analysis of clinker phase predictions using SHAP. a–c** Hierarchical ranking of clinker oxide contributions to phase predictions for alite, belite, and ferrite, respectively. Bar lengths indicate mean absolute SHAP values (wt.%), representing averaged feature impact across the test dataset. **d–f** Corresponding beeswarm plots revealing the directional influence of each oxide on phase formation. SHAP values (x-axis) indicate deviation from mean phase composition, with positive values suggesting increased formation. Color gradient (blue to red) represents oxide concentration from minimum to maximum, with point density indicating frequency of occurrence. Numbers in parentheses show oxide composition ranges (wt.%). CaO demonstrates a dominant positive correlation with alite formation, while $SiO_2$ shows a strong negative influence, aligning with established clinker chemistry.

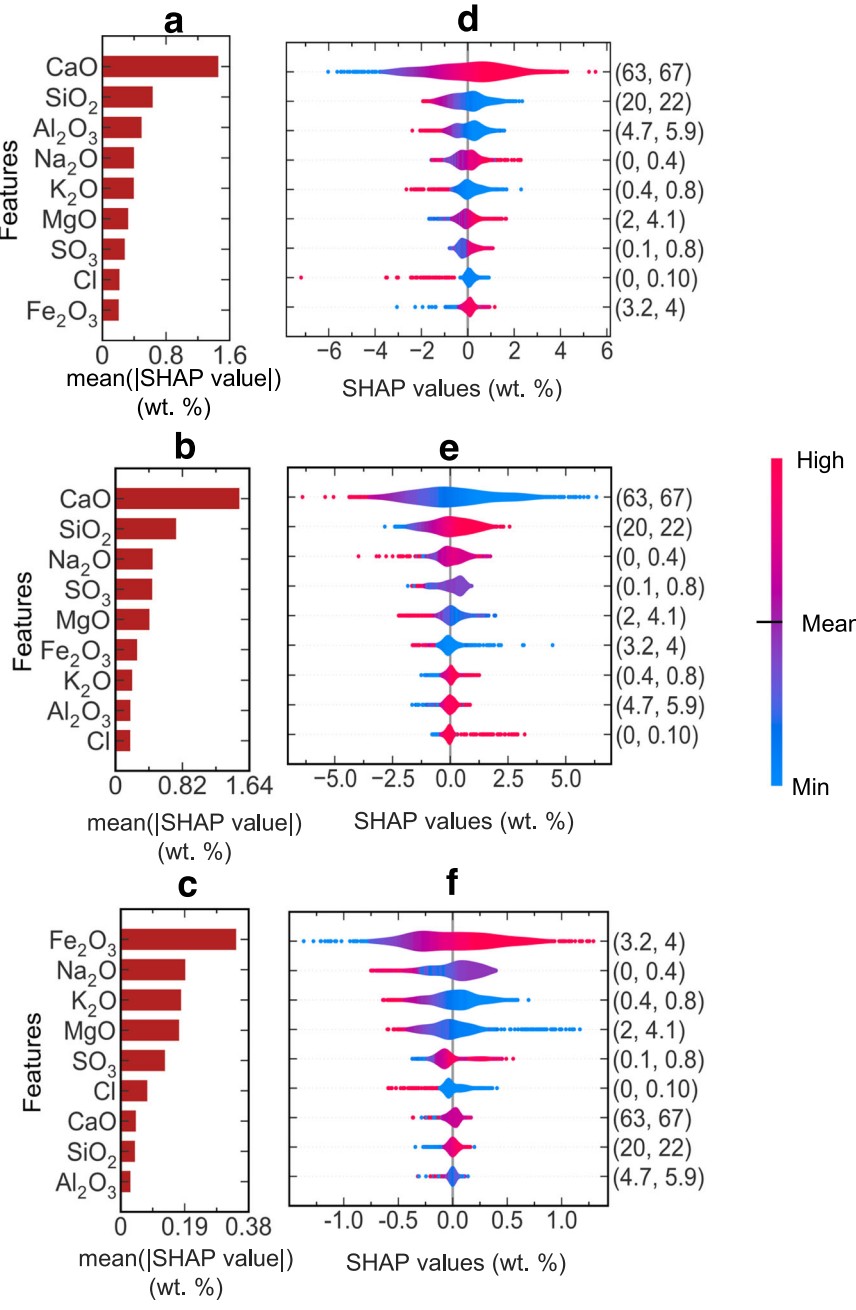

In addition to these technical concerns, industry readiness to adopt data-driven solutions and cybersecurity concerns related to sensitive process data are factors that will play a critical role in demonstrating the performance of the digital twin we aim to develop for advancing the frontiers of sustainability in cement manufacturing."

Beyond clinker prediction, this work advances several key aspects of sustainable cement manufacturing. Integrating ML-driven real-time clinker predictions with plant supervisory systems can improve energy-efficient process control. Traditional kiln temperatures are set with high safety margins to ensure complete phase formation, often leading to overburnt clinker, excessive fuel consumption, and unnecessary process emissions. Using the developed models, the operators can precisely adjust the kiln temperature and fuel dosage in real-time, minimizing fuel wastage, reliance on high-temperature operations, and over-burning related emissions, which is critical for an industry contributing ~10% of global $CO_2$ emissions. Additionally, traditional clinker quality control is based on periodic XRD lab measurements, which take hours to detect out-of-spec clinker. ML-driven real-time predictions enable early detection, allowing operators to adjust process parameters proactively and prevent defective clinker production. This considerably reduces energy-intensive reprocessing, material waste, and variability in clinker quality. Furthermore, the models developed in this work can play a crucial role in optimizing alternative fuel utilization by predicting their impact on clinker formation. By analyzing real-time fuel properties, they assess how alternative fuels influence clinker phase development, enabling operators to make precise adjustments to air-to-fuel ratios and kiln temperatures. This ensures efficient energy use while maintaining clinker quality, facilitating the transition to alternative fuels, and reducing the dependence on fossil fuels. Thus, moving towards data-driven digital twins for cement manufacturing holds the potential to improve existing industrial systems with little changes, potentially accelerating progress toward carbon-neutral manufacturing practices.

**Table 2 | Comparison of clinker composition studies described in Introduction**

| Study | Methodology | Limitation | Reported metrics for alite, belite & ferrite | Contribution from present work |
|---|---|---|---|---|
| Mastorakos et al.[14] | First principle modeling (1D dynamic CFD simulation) | • Lacks model validation | NA | • High accuracy validated on large-scale industrial data. $MAE_{alite}$: 0.75 wt.% $MAE_{belite}$: 0.96 wt.% $MAE_{ferrite}$: 0.36 wt.% |
| Hokfors et al.[21] | Physics-based modeling (process simulations using Aspen) | • Validated on just one data point from a plant • Ignores influence of process parameters | $MAE_{alite}$: 6.4 wt.% $MAE_{belite}$: 9 wt.% $MAE_{ferrite}$: 9.7 wt.% | |
| Moses et al.[22] | Statistical modeling (quadratic regression) | • Small dataset (154 data points) • Uses synthetic data; no plant validation • Models only alite; excludes belite & ferrite | $R^2_{alite}$: 1 | • Considers impact of process parameters & kiln feed • Uncertainty & extreme value testing |
| Ali et al.[30] | Machine Learning (purely data-driven neural network) | • Limited to predicting theoretical compounds ($C_3S$, $C_2S$, $C_4AF$), not actual clinker phases • Lacks model interpretability • Lacks uncertainty & rare event analysis • No benchmarking across ML models | NA | • Post hoc model explanation • Plant-specific clinker equations • Comprehensive ML benchmarking |

**Table 3 | DB1 and DB2 details: data collection frequency, measurement points, measurement techniques, and the data size**

| DB | Description | Data collection frequency | Measurement Point | Measurement Technique | Number of datapoints |
|---|---|---|---|---|---|
| DB1 | Process parameters | 1 minute | Sensor measurements at key process stages | Online measurement | 1,052,567 |
| DB2 | Composition database | 1 hour (for clinker) | Clinker cooler outlet (clinker) | XRD/XRF | 14,985 |
| | | 1 hour (for KF) | Before PHT (KF) | XRD/XRF | 15,331 |
| | | 2 hours (for HM) | Calciner outlet (HM) | XRD/XRF | 7621 |

## Methods

### Data collection

A comprehensive two-year (01/01/2020 to 31/12/2021) operational dataset collected from an industrial cement plant facilitated through the Innovandi consortium formed the foundation for our ML framework. Figure 1a illustrates the plant schematic and data collection points, with material compositions determined through XRD and XRF analyses. PP comprised 34 features, while KF, HM, and clinker comprised 9, 7, and 12 features, respectively. Thus, the total dataset consists of potentially 59 input features with the three clinker phases as the output. Complete details of the features in DB1 and DB2 are reported in Tables S1 and S2 in Supplementary A.

The database architecture comprised three distinct components:

- **DB0**: Plant configuration parameters including kiln specifications, pre-calciner characteristics, preheater architecture (strings and stages), and bypass systems.
- **DB1**: Process parameters including stage-wise temperature and pressure profiles, $O_2$ content, kiln feed temperature, and calciner fuel consumption rates. Complete specifications are provided in Table S1, Supplementary A.
- **DB2**: Compositional analyses of kiln feed (KF), hot meal (HM), and clinker, including oxide distributions and phase compositions (Table S2, Supplementary A). All compositions are reported as weight percentages (wt.%). The temporal architecture of DB1 and DB2 is detailed in Table 3

Mineral oxide compositions in KF, HM, and clinker were quantified using X-ray fluorescence (XRF) spectroscopy. The percentage of clinker phases was determined through X-ray diffraction (XRD) analysis. The list of 34 process parameters that were considered for this study are reported below in Table S1, Supplementary A. The description of composition data reported in DB2 is also shown in Table S2, Supplementary A. It is important to mention that the XRD measurements for aluminate, freelime and other minor phases were not available in the data collected from the plant. The alite and belite compositions reported in Table S2, Supplementary A are the sum of constituent polymorphs.

### Data pre-processing

The two-year operational dataset presented unique challenges characteristic of industrial-scale data collection. Measurement uncertainties stemming from instrumental limitations and human factors necessitated rigorous preprocessing to ensure data integrity. The raw dataset exhibited three primary irregularities: duplicate entries, missing measurements, and physically inconsistent values (e.g., non-normalized XRD measurements, out-of-range variables, XRF-XRD mismatches). The raw dataset reported a total of 14,985 clinker compositions.

We implemented a systematic three-tier preprocessing protocol:

1. Data Completeness check: Initial screening eliminated 207 clinker compositions lacking corresponding KF, HM, and process parameter measurements.
2. Data Consistency check:
   Consolidated duplicate entries
   Removed incomplete feature sets: 3170 entries with duplicate measurements or incomplete features were identified and removed.
   Applied 0.01-99.99 percentile filtering to exclude non-representative outliers and plant shutdown periods
3. Physical Validation: The raw dataset indicates 5857 negative compositions (see Table S2, Supplementary A for minimum kiln feed Cl, hot meal $SO_3$, and clinker Cl). Consequently, 44 rows were removed due to either negative values or XRF-XRD mismatches to ensure thermodynamic consistency.

The outlier threshold selection balanced data retention with statistical significance. As evidenced in Fig. 1b–d, excluded data points ( < 50 per variable) deviated notably from the two-year compositional distributions. This filtering methodology was consistently applied across all variables to prevent spurious correlations during model training. The final curated dataset comprised 8654 clinker measurements, representing approximately 58% of the raw data. Detailed preprocessing statistics are provided in Table 4. A comprehensive representation of the preprocessed data in comparison to the raw data is included in Fig. S1 to S4, Supplementary A.

**Table 4 | Data loss through subsequent stages of preprocessing**

| Operation | Data rows removed | Total data size | |
|---|---|---|---|
| | | **Before** | **After** |
| • **Data completeness check**: eliminating clinker compositions lacking corresponding KF, HM and/or PP measurements | 207 (1.4%) | 14,985 | 14,778 |
| • **Data consistency check**: removing data rows with missing entries and consolidating duplicate entries | 3,170 (21.2%) | 14,779 | 11,608 |
| • **Physical validation**: Enforced compositional constraints to ensure thermodynamic consistency | 44 (0.3%) | 11,608 | 11,564 |
| • **Outlier removal**: eliminating data outside the range of 0.01–99.99 percentile | 2,910 (19. 4%) | 11,564 | 80,654 |

The percentage calculation for the data rows removed is relative to the size of the raw clinker composition database, i.e., 14,985.

## Temporal synchronization protocol

The temporal resolution of collected parameters varied notably (Table 3), necessitating careful preprocessing after outlier removal. Process parameters were recorded at high frequency (1-minute intervals), whereas material compositions were measured less frequently: hourly for clinker and kiln feed, and bi-hourly for hot meal (detailed distributions in Supplementary A, Figures S1 to S4). This multi-scale temporal structure, coupled with material transport dynamics through the kiln system, required a sophisticated synchronization framework to establish causal relationships between process conditions and clinker composition.

To address this challenge, we implemented a systematic temporal synchronization protocol to account for varying sampling frequencies and material transport dynamics. Identifying the process timeline and retention times at different production stages is essential for accurately mapping the relevant process parameters with corresponding clinker compositions. The sequential timeline for clinker production in the examined plant is as follows:
1. KF measurement at preheater tower inlet ($t_0$)
2. Preheater tower residence ( ~16 minutes; total time: ($t_0 + 17$) minutes)
3. Clinker cooler retention ( ~20 minutes; total time: ($t_0 + 37$) minutes)
4. Post-production sampling delay ( ~20 minutes; total time: ($t_0 + 57$) minutes)

Thus, the total duration from kiln feed introduction to clinker formation was approximately 37 minutes. While clinker composition data was timestamped based on production time, KF, hot meal (HM), and process parameters were recorded based on their measurement times. To ensure meaningful temporal correlation, all process data were standardized to 2-hour intervals using weighted temporal averaging, incorporating key residence times: 1-minute buffer post-KF measurement, 16-minute preheater residence, 20-minute clinker cooling, and 20-minute post-production sampling delay. The alignment algorithm integrated timestamps from process parameter database (DB1) and composition database (DB2) by accounting for cumulative residence times (37 minutes total). This enabled precise mapping of clinker compositions to their corresponding input conditions, ensuring accurate temporal registration across the entire production chain–from kiln feed introduction to final clinker formation.

## ML models

Four primary feature categories - process parameters (PP), kiln feed (KF), hot meal (HM), and clinker oxides (CO) - generated 15 unique input combinations (Fig. 1a). Each combination underwent evaluation through nine algorithmic architectures: linear regression, lasso, ridge, elastic net, support vector regression, random forest, XGBoost, neural networks, and Gaussian processes. Mathematical formulations of these modes are provided in Supplementary B. All the models were trained using the sklearn package.

## Performance metrics

Model assessment utilized three complementary metrics described below.
(i)  Mean Absolute Percentage Error (MAPE).[37]:

$$MAPE = \frac{1}{n}\sum_{i=1}^{n}\frac{|y_p(i) - y_t(i)|}{y_t(i)} \qquad (1)$$

where $n$ represents sample size, $y_p(i)$ and $y_t(i)$ denote predicted and true values.
(ii)  Mean Absolute Error (MAE):

$$MAE = \frac{1}{n}\sum_{i=1}^{n}|y_p(i) - y_t(i)| \qquad (2)$$

(iii)  Coefficient of Determination ($R^2$)[38]:

$$R^2 = 1 - \frac{\text{RSS}}{\text{TSS}} \qquad (3)$$

where,

$$\text{RSS} = \sum_{i=1}^{n}(y_p(i) - y_t(i))^2 \quad \text{TSS} = \sum_{i=1}^{n}(y_p(i) - \bar{y})^2 \quad \bar{y} = \frac{1}{n}\sum_{i=1}^{n}(y_p(i) - y_t(i))^2$$

## Model optimization protocol

Hyperparameter optimization followed a systematic protocol[39]:
1. Data partitioning: 80:20 train-test split maintaining statistical equivalence via sklearn[40].
2. Parameter optimization: 4-fold cross-validation[41] utilizing GridSearchCV[42].
3. Model selection: Optimization based on validation scores (Table S3, Supplementary C).
4. Validation: Final assessment on holdout test set.

Detailed optimization protocols appear in Supplementary C. The hyperparametric optimization was performed using the GridSearchCV in the sklearn package.

## Shapley Additive Explanations (SHAP)

To interpret the learned black-box functions within complex ML architectures, we implemented SHAP analysis[43], a post-hoc interpretation framework derived from game-theoretic Shapley values[44]. This methodology quantifies individual feature contributions to model predictions through systematic feature importance allocation. The SHAP value ($\chi_k$) for feature $k$ represents its average marginal contribution across all possible feature subsets[45]:

$$\chi(f, X) = \sum_{\ddot{z} \subset X'} \frac{|\ddot{z}|(n - |\ddot{z}| - 1)!}{n!} f_X - f_X(\ddot{z}_{\setminus k})] \qquad (4)$$

where, $f$: Original model architecture, $X$: Complete feature set, $|\ddot{z}|$: Cardinality of feature combinations within non-zero feature power set, $n$: Simplified input feature count, $\ddot{z} \subset X'$: Vectors with non-zero entries subset to $\ddot{X}$, $f_X(\ddot{z})$: Model prediction for given data point, $f_X(\ddot{z}_{\setminus k})$: Model prediction excluding feature $k$[46].

SHAP values correlate directly with prediction error magnitude - larger errors indicate greater feature importance. The framework accommodates various model-specific approximations (kernel, Deep, Linear, Tree-based explainers), reducing complex architectures to tractable polynomial

forms[39,47]. Comprehensive theoretical foundations can be found in[43]. SHAP library was employed to perform the SHAP computations.

## Data availability

The raw data used in this study was obtained from an industry partner under a confidentiality agreement with the Global Cement and Concrete Research Network (GCCRN). The identity of the industry partner and the raw dataset cannot be disclosed due to proprietary restrictions. Consequently, the data is not publicly available. Any inquiries regarding the data can be directed to the corresponding author, subject to approval from the industry partner and GCCRN.

## Code availability

All the source codes developed as part of this work are publicly available at the following repository: [https://doi.org/10.5281/zenodo.15302727][48].

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

## Acknowledgements

The authors acknowledge the funding and data collection support by Innovandi - The Global Cement and Concrete Research Network (GCCRN). The authors also acknowledge the high-performance computing (HPC) platform at the Indian Institute of Technology Delhi for the computational and storage resources.

## Author contributions

Sheikh Junaid Fayaz: Conceptualization, data curation, formal analysis, investigation, methodology, software implementation, validation, visualization, writing - original draft, and writing - review and editing. Nestor Montiel-Bohorquez: Conceptualization, investigation, validation, and writing - review and editing. Shashank Bishnoi: Conceptualization, data curation, funding acquisition, investigation, project administration, supervision, validation, and writing - review and editing. Matteo Romano: Conceptualization, data curation, funding acquisition, investigation, project administration, supervision, validation, and writing - review and editing. Manuele Gatti: Conceptualization, data curation, funding acquisition, investigation, project administration, supervision, validation, and writing - review and editing. N. M. Anoop Krishnan: Conceptualization, data curation, formal analysis, funding acquisition, investigation, methodology, project administration, resources acquisition, supervision, validation, writing - original draft, and writing - review and editing.

## Competing interests

The authors declare no competing interests.

## Ethical approval

This research was conducted with the approval of Cement and Concrete Research Network (GCCRN), the consortium supporting the research through funding and facilitation of data acquisition. All data utilized in this study were anonymized prior to analysis to ensure that no sensitive or proprietary information related to the participating cement plants was disclosed.
