## [Transparent Peer Review file · Communications Engineering]

Industrial-scale Prediction of Cement Clinker Phases using Machine Learning

Corresponding Author: Professor N M Anoop Krishnan

Version 0:

Reviewer comments:

Reviewer #1

(Remarks to the Author)

1- The abstract is too general; the authors should include the final numerical results.

2- The introduction needs to improve and the authors need to extend the literature review and use more papers and book chapters related to this title, also they should mention about weak points of other papers and the reason they use this idea.

3- After the introduction, the Authors need to add background study about models similar to machine learning.

4—The Cement plant section can not be added to the results part. It would be good if the Authors added a section about the dataset and all information about it, using a correlation matrix, P-P diagram, Box normal plot, etc.

5- In Table 2 the authors used many models of ML but I know each model has many characteristics that affect the results, for example in NN or ANN, there are many types of NN including feed-forward, MLP, RBF, and each model has different topology and behavioral performance, the authors did not mention to details.

About SVM similar to NN many variables affected the results.

It is better for the authors to explain each model in detail or select the top three models and explain more.

6- I strongly suggest the authors compare the results with other papers and literature reviews.

7- The R-square for three outputs is 0.90 for NN, 0.8 for GPR, and 0.7 for SVR, please explain more the authors: use one model with three outputs or three models, and each model has one output.

8- The authors should add a part about cement clinker Phases and the advantages and disadvantages.

Reviewer #2

(Remarks to the Author)

The paper demonstrates that machine learning can accurately predict clinker phases (alite, belite, ferrite) in real-time, outperforming traditional models like the Bogue equation. Using a two-year dataset from an industrial cement plant, the work introduces a novel, large-scale machine learning approach for cement manufacturing, with the potential to reduce material waste, improve quality control, and support sustainable practices. This approach is highly relevant for both the cement industry and industrial process optimization.

The results clearly show that machine learning models, such as neural networks and Gaussian process regression, outperform the Bogue equation in predicting clinker compositions. The paper is convincing, with solid statistical methods (MAPE, MAE, R^2) and rigorous validation. However, a deeper discussion on model limitations, particularly in handling extreme operational conditions and data anomalies, would strengthen the paper. Additionally, more out-of-sample testing and uncertainty analysis would improve model robustness.

The paper's findings could influence both cement manufacturing and broader machine learning applications in industrial control. It provides a promising direction for reducing CO₂ emissions and improving efficiency. While reproducible to some extent through the provided GitHub code, full reproducibility depends on access to proprietary data from the cement plant.

Recommendations:

- Expand on preprocessing steps and data synchronization challenges.
- Include more out-of-sample validation to strengthen model generalizability.
- Consider practical deployment strategies for real-world industrial use.

Reviewer #3

(Remarks to the Author)

This paper presents a ML-based approach for predicting clinker mineralogy in an industrial cement plant using a two-year operational dataset. The study also discusses the potential implementation of the proposed framework as a digital twin for cement production. The paper is well-structured, and its findings are relevant for improving the efficiency of cement manufacturing. However, some areas require further clarification and improvement before it is suitable for publication in this journal.

- Line 6: The first sentence of the Abstract should state "2.4 billion tonnes" instead of "2.4 tonnes" to correct the factual inaccuracy.

General comment on Abstract: The abstract should be more informative and quantitative, explicitly mentioning the ML models developed and key results obtained in the study.

- Line 23: The correct average CO₂ emission per tonne of cement is ~0.8 tonne, not ~0.6 tonne (as widely reported in the literature). Additionally, this statement requires a supporting reference.

- The paper refers to the ML framework as a "digital twin", but it lacks key characteristics of a fully developed digital twin. A true digital twin involves real-time feedback loops, process control, and dynamic simulations of the physical system. Since the study primarily focuses on predictive modeling without automated adjustments, the authors should clarify that their approach is a data-driven digital representation rather than a fully developed digital twin.

- While the paper claims that the ML model enables real-time clinker prediction, it does not discuss how the model would be deployed in an industrial setting. Real-time deployment requires considerations such as computational efficiency, integration with existing control systems, and handling of sensor noise and data delays.

- The study utilises 59 input features, but there is no discussion on feature selection or the minimum set of variables required for accurate predictions. A feature importance analysis would enhance the study's practical applicability.

- Given the cement industry's significant contribution to global CO₂ emissions, the study would benefit from a discussion on how ML-based clinker optimization can support sustainability goals (e.g., energy efficiency, alternative fuels, emissions reduction).

- The conclusion section should include a dedicated paragraph on future research directions. Additionally, while the paper compares ML-based predictions with the traditional Bogue equation, it does not consider first-principle physics-based models, which are widely used in modern cement manufacturing for process optimization. Including a comparison with a physics-based model would provide a more comprehensive assessment of the ML model's performance.

Version 1:

Reviewer comments:

Reviewer #1

(Remarks to the Author)

Dear Editor

Following the examination of the revised manuscript together with the response letter, it is confirmed that the authors responded to all reviewer comments effectively. Accordingly, it is advised to accept the article titled: "Industrial-scale Prediction of Cement Clinker Phases using Machine Learning" in the journal.

Best Regards

Reviewer #2

(Remarks to the Author)

The authors have thoroughly addressed the review comments, and the improvements made to the paper, particularly in terms of model transparency and practical deployment strategies, have strengthened the overall contribution. The application of machine learning to predict clinker phases at an industrial scale is both novel and highly relevant to the cement industry and industrial process optimization. The statistical analysis and validation against traditional methods demonstrate the robustness and accuracy of the models. While further exploration of rare event scenarios and integration with real-time plant control systems would further enhance the paper, the work is already a valuable contribution to the field. Given the improvements made by the authors and the strong technical and practical contribution of the paper, I recommend accepting the paper.

Dear Editor and Reviewers,

Thank you for considering our manuscript and for the careful review. A detailed, point-by-point response to all the recommendations and comments from each of the reviewers is included below, with reviewers' comments marked in black and our responses to the comments marked in blue. Additionally, a revised manuscript with all changes clearly marked in blue is also included. We believe that the revision has significantly improved the manuscript, and with these changes, we hope that you will now find the manuscript suitable for publication in *Communications Engineering*.

Reviewer 1:

Thank you for the positive comments on the manuscript. Please find the responses below.

Comments to the Authors:

1. The abstract is too general; the authors should include the final numerical results.

Response: Thank you. We have now revised the abstract with numerical results and the effectiveness of our approach compared to traditional Bogue equations.

To address this comment, we have now modified the abstract as follows:

“Cement production exceeds 4.1 billion tonnes annually, emitting 2.4 billion tonnes of CO₂ annually, necessitating improved process control. Traditional models, limited to steady-state conditions, lack predictive accuracy for clinker mineralogical phases. Here, using a comprehensive two-year industrial dataset, we develop machine learning models that significantly outperform conventional Bogue equations with mean absolute percentage errors of 1.24%, 6.77%, and 2.53% for alite, belite, and ferrite prediction respectively, compared to 7.79%, 22.68%, and 24.54% for Bogue calculations. Our models remain robust under varying operations and are evaluated for uncertainty and rare-event scenarios. Through post hoc explainable algorithms, we interpret the hierarchical relationships between clinker oxides and phase formation, providing insights into the functioning of an otherwise black-box model. The framework can potentially enable real-time optimization of cement production, thereby providing a route toward reducing material waste and ensuring quality while reducing the associated emissions under real-world conditions.”

2. The introduction needs to improve, and the authors need to extend the literature review and use more papers and book chapters related to this title, also they should mention about weak points of other papers and the reason they use this idea.

Response: We appreciate the reviewer's suggestion. The introduction has been thoroughly revised to provide a more comprehensive background. The revised section now includes:

- A general overview of the problem,
- The current status of traditional quality control in the industry,
- Challenges associated with first-principle models,
- Limitations of physics-based models,
- Recent developments in machine learning for cement production,
- The motivation and necessity for this study.

Additionally, we have extended the literature review by incorporating more relevant studies. We have also discussed the weaknesses of existing studies and justified our approach in

response to these limitations in Discussion section. We hope the revised introduction now meets the reviewer's expectations.

3. After the introduction, the Authors need to add background study about models similar to machine learning.

Response: We appreciate the reviewer's suggestion. A comprehensive background study on all relevant models for clinker phase prediction has been included. These models have been thoroughly discussed and compared in detail within the introduction and in *Table 2* of the *Discussion section*.

4. The Cement plant section cannot be added to the results part. It would be good if the Authors added a section about the dataset and all information about it, using a correlation matrix, P-P diagram, Box normal plot, etc.

Response: We appreciate the reviewer's comment. We have added a dedicated sections in the *Methods* to comprehensively describe dataset details: *4.1 Data Collection*, *4.2 Data Preprocessing*, and *4.3 Temporal Synchronization Protocol*. Given the journal's constraint of a maximum 10 display items in the main text, we could not incorporate additional figures or tables on data exploration within the main text. However, extensive visualization and statistical analysis of the dataset is provided in **Appendix A**:

- *Table A5* and *Table A6* detail the key statistical properties (mean, minimum, maximum, standard deviation, range, and units) for the 63 features used in the study.

- *Figures A7–A10* present scatter plots and histograms, illustrating feature distributions and trends in temporal evolution of each parameter, distributions of raw vs. filtered data.

- In response to the reviewer's suggestion, we have now included *Figure A11* in *Appendix A*, which visualizes feature interactions across the process parameter (PP), kiln feed (KF), hot meal (HM), and clinker oxides (CO) databases, in relation to the clinker phase predictions.

“Figure A11: Heatmaps showing Pearson's correlation for (a) PP, (b) KF, (c) HM, (d) CO”

Regarding the placement of preprocessed-dataset discussion in the results, we acknowledge the reviewer's concern. However, since this analyzing the preprocessed data is integral to the model development process and does not naturally fit within the Methods section, we have included it in the Results section as the most appropriate placement within the journal's format. We hope that this addresses the reviewer's concern and enhance the clarity and completeness of our dataset presentation.

- In Table 2 the authors used many models of ML but I know each model has many characteristics that affect the results, for example in NN or ANN, there are many types of NN including feed-forward, MLP, RBF, and each model has different topology and behavioural performance, the authors did not mention to details. About SVM similar to NN many variables affected the results.

It is better for the authors to explain each model in detail or select the top three models and explain more.

Response: We appreciate reviewer's concern regarding model details and fully agree that model topology significantly impacts results. To address this, we are already providing a comprehensive explanation of all nine ML models used in the study, as highlighted below:

- **Mathematical Formulation:** Detailed mathematical formulation of each model is provided in **Appendix B** for readers interested in the theoretical and architectural nuances.
- **Model Topology & Hyperparameters:** The topology of each model, including optimized hyperparameters, is presented in **Table C7 (Appendix C)**. Additionally, the impact of the optimized hyperparameters on model performance is analysed in **Figures C12–C15 (Appendix C)**.
- **SVR, GPR and NN type:** Kernel function for SVR and GPR is already mentioned and can be found at lines 595 and 612 in the revised manuscript and further detailed in **Table C7 (Appendix C)**. To clarify the type of NN used in the study, we have added the following text (see line 620 in revised manuscript) in **Appendix B.8** where we explain the NN architecture:

“In this study, we employ a fully connected feedforward neural network consisting of an input layer, multiple hidden layers, and an output layer.”

Due to space constraints we could not include these details in the main text. However, the appendices are referenced throughout the text where necessary. We hope this addresses the reviewer's comment. Please note that Table 2 mentioned in the comment will be Table 1 in the revised manuscript.

6. I strongly suggest the authors compare the results with other papers and literature reviews.

Response: We appreciate the reviewer's suggestion. A detailed comparison with relevant studies has been included in the *Introduction* highlighting their methodologies, findings and limitations (see line 22-47 in revised manuscript). Additionally, specific numerical results have been compared in *Table 2*. The added table is given below:

Table 2: Comparison of clinker composition studies in the Introduction¹

Study	Methodology	Limitation	Reported metrics for alite, belite & ferrite	Contribution from present work
Mastorakos et al. [14]	First principle modeling (1D dynamic CFD simulation)	 Lacks model validation 	NA	 Unprecedented accuracy validated on large-scale industrial data. MAE_{alite}: 0.75 wt. % MAE_{belite}: 0.96 wt. % $MAE_{ferrite}$: 0.36 wt. %
Hokfors et al. [21]	Physics-based modeling (process simulations using Aspen)	 Validated on just one data point from a plant Ignores influence of process parameters 	MAE_{alite} : 6.4 wt. % MAE_{belite} : 9 wt. % $MAE_{ferrite}$: 9.7 wt. %	
Moses et al. [22]	Statistical modeling (quadratic regression)	 Small dataset (154 data points) Uses synthetic data; no plant validation Models only alite; excludes belite & ferrite 	R^2_{alite} : 1	 Considers impact of process parameters & kiln feed Uncertainty & extreme value testing
Ali et al. [30]	Machine Learning (purely data-driven neural network)	 Limited to predicting theoretical compounds (C_3S, C_2S, C_4AF), not actual clinker phases Lacks model interpretability Lacks uncertainty & rare event analysis No benchmarking across ML models 	NA	 Post hoc model explanation Plant-specific clinker equations Comprehensive ML benchmarking

7. The R-square for three outputs is 0.90 for NN, 0.8 for GPR, and 0.7 for SVR, please explain more the authors: use one model with three outputs or three models, and each model has one output.

Response: We appreciate the reviewer’s comment. In our study, we use separate models to predict each clinker phase rather than a single multi-output model. We employ individual models for predicting alite, belite, and ferrite. The R^2 values referred in the comment correspond to the best-performing models for each phase: NN achieves highest accuracy for alite prediction ($R^2 = 0.90$), GPR for belite ($R^2 = 0.80$), and SVR for ferrite ($R^2 = 0.70$).

Since the underlying correlation between input parameters and each phase varies, having a single model for all phases would be suboptimal. Also, using single output models allows us to tune hyper-parameters of each model specifically for its target phase, improving generalization and prediction fidelity for each phase.

To clarify the same to the readers, we have added the following text in the revised manuscript at line no 99 in the revised manuscript:

“Note that, in the study separate models have been used to predict each clinker phase rather than multi-output models. This allows independent hyper-parameter tuning of each model, optimizing its performance for the specific target phase.”

8. The authors should add a part about cement clinker phases and the advantages and disadvantages.

Response: A section has been added (see line 10 in revised manuscript) as given below:

“The performance of cement, particularly its 28-day compressive strength, is primarily governed by the clinker’s mineralogical phases—alite, belite, aluminate, and ferrite. Alite drives early strength development, belite contributes to long-term strength¹, and ferrite influences color and provides minor early-age strength. However, while alite-rich cements enhance early strength, alite content > 65%² can lead to increased heat of hydration and increased CO₂ emissions due to higher limestone requirements in the raw feed. Conversely, belite-rich cements exhibit improved long-term durability but may delay early strength gain due to lower reactivity. Therefore, determining the relative composition of these phases is critical in determining the cement quality³.”

Reviewer 2:

The paper demonstrates that machine learning can accurately predict clinker phases (alite, belite, ferrite) in real-time, outperforming traditional models like the Bogue equation. Using a two-year dataset from an industrial cement plant, the work introduces a novel, large-scale machine learning approach for cement manufacturing, with the potential to reduce material waste, improve quality control, and support sustainable practices. This approach is highly relevant for both the cement industry and industrial process optimization.

The results clearly show that machine learning models, such as neural networks and Gaussian process regression, outperform the Bogue equation in predicting clinker compositions. The paper is convincing, with solid statistical methods (MAPE, MAE, R^2) and rigorous validation. However, a deeper discussion on model limitations, particularly in handling extreme operational conditions and data anomalies, would strengthen the paper. Additionally, more out-of-sample testing and uncertainty analysis would improve model robustness.

The paper's findings could influence both cement manufacturing and broader machine learning applications in industrial control. It provides a promising direction for reducing CO₂ emissions and improving efficiency. While reproducible to some extent through the provided GitHub code, full reproducibility depends on access to proprietary data from the cement plant.

Response: Thank you for the positive comments.

We would like to clarify that both uncertainty analysis and out-of-sample testing is included in the work using the available plant data. This is now clearly explained in the revised manuscript as follows.

“ML model robustness was evaluated through 20 independent training iterations using different random seeds. Figure 2 presents the mean predictions obtained from these 20 trained models. The gray bands enveloping the mean predictions in Fig. 2 e.g,i represent the uncertainty in model predictions arising from variations across the 20 training runs. Performance of the best-performing models, along with error bars representing uncertainty, is shown in Fig. C.16.”

“Figure C.16: Performance of best performing models: (a) NN for alite, (b) GPR for belite, (c) SVR for ferrite) using all the 59 input features. Black circles represent the mean model prediction, while red error bars indicate model uncertainty ($\pm 3\sigma$)”

Additionally, the comment on OOS testing and model limitation has been addressed in detail in Recommendation 2. The pointwise responses to specific recommendations are given below.

Recommendation 1: Expand on preprocessing steps and data synchronization challenges.

Response: Subsection 4.2 *Data pre-processing* in *Methods* has been revised to expand on the preprocessing steps. Additionally, we have added a new subsection 4.3 *Temporal synchronization protocol* has been added in *Methods* to elaborate on the synchronization challenges.

Recommendation 2: Include more out-of-sample (OOS) validation to strengthen model generalizability.

Response: We appreciate the reviewer’s concern regarding model robustness. Our models are tested on a completely unseen test dataset, ensuring that none of the test points were included in the training process. Thus, the model is already evaluated on out-of-sample (OOS) data. As shown in Figure 1 (e-g), the statistical properties of the training and test sets are similar. Please note that this data is still not out-of-distribution as the data is taken from the same plant albeit for a different “time period” than those considered for training the model. In other words, the data is unseen and out-of-sample, but still not out-of-distribution.

To further assess the model’s performance under rare extreme event scenarios, we have added a new section 2.3 *Model limitations: Extreme event scenarios* (see line 168 – 178 in revised manuscript) as given below:

“Model limitations: Rare event scenarios: While the ML models demonstrate remarkable accuracy in predicting clinker phase compositions, it is crucial to assess their limitations, particularly their performance under extreme, rare plant operating conditions with very low occurrence frequency. The models were trained on a preprocessed dataset, free from anomalies and extreme values, as outliers beyond the 0.01–99.99 percentile range were removed. As shown in Fig. 5(a-c), the number of phase composition data points outside this filtering window is minimal (26 for alite, 104 for belite, and 58 for ferrite). Fig. 5(d-e) further illustrates that the probability of these rare compositions is below 1%, with the plant operating within the normal range for over 99% of the time. However, when tested on these rare events, the models exhibit significant deviations, leading to a noticeable spike in prediction errors, as shown in Fig.5(g-h). This highlights a fundamental limitation of data-driven models—their inability to extrapolate beyond the training range. Nonetheless, given that the plant operates within the normal range for the vast majority of the time, the developed ML models remain highly effective for quality control and process optimization in routine operations.”

Figure 5: **Extreme event model evaluation.** Distribution of (a) alite, (b) belite, and (c) ferrite over two years of plant operation. Data retained after preprocessing is shown in green, while data removed during the three-tier outlier removal process is shown in red. The red-marked events correspond to rare plant operations with very low occurrence frequency, denoted as f for each phase. (d-f) illustrate the performance of the best-performing models across normal operation ranges (green background) and rare operation conditions (red background). The total probability of normal and rare operations is also indicated for each phase. The prediction errors for (g) normal operation and (h) extreme operation are shown using box plots of absolute errors.

We hope this addition addresses the concern on OOS validation raised by the reviewer.

Recommendation 3: Consider practical deployment strategies for real-world industrial use.

Response: The deployment strategies for real world industrial use has been outlined in the Discussion (see line 210-228 in revised manuscript). The added text is as follows:

- **“Latency due to Model fine-tuning:** As plant operations can evolve beyond the range of the training data used in the models, continuous fine-tuning with new operational data would be necessary to ensure efficient adaptation to evolving plant operation. Various parallelization techniques can be evaluated on GPUs to accelerate fine-tuning time. Also, to minimize cloud-based latency, models will be deployed on local industrial PCs or edge devices near kiln control systems.
- **Integrating models with existing Control Systems** requires: (a) developing an API to enable real-time communication between the ML model and control systems, ensuring

structured model outputs compatible with logic controllers managing kiln processes; (b) implementing real-time dashboards for operators to monitor predictions, receive alerts, and take corrective actions; and (c) establishing fail-safe mechanisms with fallback logic that reverts to traditional control methods if the ML model encounters issues, ensuring uninterrupted plant operations.

- ***Handling Sensor Noise and Data Delays:*** *real-time outlier detection and filtering pipelines will be implemented to manage noisy sensor inputs. For temporary sensor failures, interpolation, last-known values, or suitable imputation methods will be used. Time synchronization, based on process delays, will ensure proper alignment of sensor data collected from different process stages.*

In addition to these technical concerns, industry readiness to adopt the data-driven solutions and cybersecurity concerns related to sensitive process data are factors that will play a critical role in demonstrating the performance of the digital-twin we aim to develop for advancing the frontiers of sustainability in cement manufacturing.”

Reviewer 3:

This paper presents a ML-based approach for predicting clinker mineralogy in an industrial cement plant using a two-year operational dataset. The study also discusses the potential implementation of the proposed framework as a digital twin for cement production. The paper is well-structured, and its findings are relevant for improving the efficiency of cement manufacturing. However, some areas require further clarification and improvement before it is suitable for publication in this journal.

Response: Thank you for the positive comments. The pointwise responses to specific comments are given below.

1. Line 6: The first sentence of the Abstract should state “2.4 billion tonnes” instead of “2.4 tonnes” to correct the factual inaccuracy.

Response: Thank you for pointing this out. We have corrected it to 2.4 billion tonnes in the revised abstract.

2. General comment on Abstract: The abstract should be more informative and quantitative, explicitly mentioning the ML models developed and key results obtained in the study.

Response: We thank the reviewer for the comment.

To address this comment, we have now modified the abstract as follows:

“Cement production exceeds 4.1 billion tonnes annually, emitting 2.4 billion tonnes of CO₂ annually, necessitating improved process control. Traditional models, limited to steady-state conditions, lack predictive accuracy for clinker mineralogical phases. Here, using a comprehensive two-year industrial dataset, we develop machine learning models that significantly outperform conventional Bogue equations with mean absolute percentage errors of 1.24%, 6.77%, and 2.53% for alite, belite, and ferrite prediction respectively, compared to 7.79%, 22.68%, and 24.54% for Bogue calculations. Our models remain robust under varying operations and are evaluated for uncertainty and rare-event scenarios. Through post hoc explainable algorithms, we interpret the hierarchical relationships between clinker oxides and phase formation, providing insights into the functioning of an otherwise black-box model. The framework can potentially enable real-time optimization of cement production, thereby providing a route toward reducing material waste and ensuring quality while reducing the associated emissions under real-world conditions.”

3. Line 23: The correct average CO₂ emission per tonne of cement is ~0.8 tonne, not ~0.6 tonne. Additionally, this statement requires a supporting reference.

Response: We appreciate the reviewer’s concern on factual accuracy. Based on the literature we find different emission factor for cement production ranging from 0.55 to 0.879 tonne CO₂/tonne cement^{4,5}. The numbers differ as explained in a study⁶ wherein the authors assemble large variety of available datasets on CO₂ emission from cement production. The authors conclude that owing to different clinker ratio, fuel type, energy efficiency of the plant, raw material composition and regional electricity mix to the plant, the emission factor can vary instead of having a fixed value under all conditions.

The value of 0.66 tonne CO₂/tonne of cement has been used in various studies^{4,7}. We have now added the supporting reference for the value in the revised manuscript (see line 8). Thank you for pointing out the missing reference.

4. The paper refers to the ML framework as a “digital twin”, but it lacks key characteristics of a fully developed digital twin. A true digital twin involves real-time feedback loops, process control, and dynamic simulations of the physical system. Since the study primarily focuses on predictive modeling without automated adjustments, the authors should clarify that their approach is a data-driven digital representation rather than a fully developed digital twin.

Response: We agree with the reviewer and acknowledge that our current work does not fully meet the criteria of a digital twin, as it lacks real-time feedback loops, automated process control, and dynamic simulations of the physical system. Ensuring correctness in terminology, we have revised the manuscript to ensure the present work is not described as a digital twin.

Furthermore, while our current study primarily focuses on predictive modelling, the developed ML models have the potential to be integrated into an ML-based digital twin for cement production. Hence, as part of future research, we plan to extend this work by incorporating real-time data streams and adaptive process control mechanisms to bridge the gap toward a fully functional digital twin.

We have outlined the pending work as a part of future work that is required to advance the present work to a fully functional digital twin in the Discussion section from line 210-228 in the revised manuscript.

5. While the paper claims that the ML model enables real-time clinker prediction, it does not discuss how the model would be deployed in an industrial setting. Real-time deployment requires considerations such as computational efficiency, integration with existing control systems, and handling of sensor noise and data delays.

Response: The deployment challenges have been added in the revised manuscript (line 210-228) as given below:

“While this study focuses on predictive modeling of clinker mineralogy, deploying these models in industrial settings is crucial for realizing their broader implications in advancing sustainability goals. However, plant demonstration presents several challenges, outlined below as future research directions.

- **Latency due to Model fine-tuning:** *As plant operations evolve beyond the training range of the models, continuous fine-tuning with new operational data would be necessary to ensure efficient adaptation to evolving plant operation. Various parallelization techniques can be evaluated on GPUs to accelerate fine-tuning time. Also, to minimize cloud-based latency, models will be deployed on local industrial PCs or edge devices near kiln control systems.*
- **Integrating models with existing Control Systems** *requires: (a) developing an API to enable real-time communication between the ML model and control systems, ensuring structured model outputs compatible with logic controllers managing kiln processes; (b) implementing real-time dashboards for operators to monitor predictions, receive alerts, and take corrective actions; and (c) establishing fail-safe mechanisms with fallback logic that reverts to traditional control methods if the ML model encounters issues, ensuring uninterrupted plant operations.*

- **Handling Sensor Noise and Data Delays:** *real-time outlier detection and filtering pipelines will be implemented to manage noisy sensor inputs. For temporary sensor failures, interpolation, last-known values, or suitable imputation methods will be used. Time synchronization, based on process delays, will ensure proper alignment of sensor data collected from different process stages.*

In addition to these technical concerns, industry readiness to adopt the data-driven solutions and cybersecurity concerns related to sensitive process data are factors that will play a critical role in demonstrating the performance of the digital-twin we aim to develop for advancing the frontiers of sustainability in cement manufacturing.”

6. The study utilises 59 input features, but there is no discussion on feature selection or the minimum set of variables required for accurate predictions. A feature importance analysis would enhance the study’s practical applicability.

Response: Thank you for the great suggestion. We agree with the reviewer that it might be challenging to 59 input features, including the hot meal measurements, in a realistic setting. Also, given that all the 34 process parameters (PP) are not independent, rather 24 PP are dependent on the 10 independent PP in a plant we have added a section on sparse ML models discussing minimum set of PP required for prediction with less than 12% error margin. The added text is as follows (see line 157-167 in revised manuscript):

“**Sparse ML models:** While maximal-information ML models leveraging all 59 input features— including process PP, KF, HM, and CO —demonstrate exceptional predictive accuracy for clinker phase composition, their practical implementation could be challenging. Many process parameters exhibit strong correlations (see Fig. A11) and obtaining a complete dataset with all 59 features in an industrial setting may not always be feasible. To address this, we propose parsimonious ML models which can predict alite, belite, and ferrite phases with MAPE of 2.91%, 11.41%, and 3.22%, respectively (Fig. 4, b-d). The models rely on a minimal number of inputs, utilizing only 10 independently controllable PP (marked with * in Table A5) along with KF composition. The selected 10 PP are independent and exhibit low mutual correlation (see Fig A11), while the remaining 24 process parameters are dependent variables, determined as a consequence of plant operations governed by these 10 controllable inputs. Despite using a reduced input set, the parsimonious ML model maintains high predictive accuracy, making it both practical and interpretable. While slightly less accurate than maximal-information models, it significantly outperforms Bogue and the developed clinker equations, demonstrating ML's potential in data-limited industrial settings.”

“Figure 4: **Input feature pruning:** a, MAPE of optimal machine learning models (Neural Network for alite, Gaussian Process Regression for belite, Support Vector Regression for ferrite) across 15 combinations of input features: process parameters (PP), kiln feed (KF), hot meal (HM), and clinker oxides (CO). Values in parentheses represent MAPE (%) for alite (green), ferrite (blue), and belite (red) predictions. **Sparse ML models for clinker phase prediction:** (b) Alite prediction using NN, (c) Belite prediction using GPR, and (d) Ferrite prediction using SVR, utilizing 10 controllable PP and KF compositions. The top-right inset shows each phase’s best-performing model—NN for alite, GPR for belite, and SVR for ferrite—when trained with all 59 input features. Black circles represent the mean model prediction, while red error bars indicate model uncertainty $\pm 3\sigma$. The bottom-left inset presents a comparative analysis of MAPE for the sparse model against the best-performing model, as well as the Bogue and clinker equations developed in this study.”

“Figure A11: Heatmaps showing Pearson’s correlation for (a) PP, (b) KF, (c) HM, (d) CO”

- Given the cement industry’s significant contribution to global CO₂ emissions, the study would benefit from a discussion on how ML-based clinker optimization can support sustainability goals (e.g., energy efficiency, alternative fuels, emissions reduction).

Response: A discussion elaborating the role of ML-based optimization towards achieving sustainability goals has been incorporated in the revised manuscript (see line no 229-242). The added text is as follows.

“Beyond clinker prediction, this work advances several key aspects of sustainable cement manufacturing. Integrating ML-driven real-time clinker predictions with supervisory systems at plant can enhance energy-efficient process control. Traditional kiln temperatures are set with high safety margins to ensure complete phase formation, often leading to overburnt clinker, excessive fuel consumption, and unnecessary process emissions. Using the developed models, the operators can precisely adjust the kiln temperature and fuel dosage in real-time, minimizing fuel wastage, reliance on high-temperature operations and over-burning related emissions, which is critical for an industry contributing ~10% of global CO₂ emissions. Additionally, traditional clinker quality control relies on periodic XRD lab measurements,

which take hours to detect out-of-spec clinker. ML-driven real-time predictions enable early detection, allowing operators to adjust process parameters proactively and prevent defective clinker production. This significantly reduces energy-intensive reprocessing, material waste, and variability in clinker quality. Furthermore, the developed models play a crucial role in optimizing alternative fuel utilization by predicting their impact on clinker formation. By analyzing real-time fuel properties, they assess how alternative fuels influence clinker phase development, enabling operators to make precise adjustments to air-to-fuel ratios and kiln temperatures. This ensures efficient energy use while maintaining clinker quality, facilitating the transition to alternative fuels and reducing dependence on fossil fuels “

- The conclusion section should include a dedicated paragraph on future research directions. Additionally, while the paper compares ML-based predictions with the traditional Bogue equation, it does not consider first-principle physics-based models, which are widely used in modern cement manufacturing for process optimization. Including a comparison with a physics-based model would provide a more comprehensive assessment of the ML model’s performance.

Response 8: We appreciate the suggestion. We have added a paragraph on future research directions in the *Discussion* section of the revised manuscript (see line 210-228). The future work focuses on the model deployment strategies as already discussed in comment 5.

Regarding the reviewer’s suggestion to compare our model with first-principle and physics-based models, we have added a detailed comparison with first principle and physics based models in the *Introduction* (see line 22-35 in the revised manuscript). Also, the summary of the comparison is presented in *Table 2* in the *Discussion* as well as given below:

Table 2: Comparison of clinker composition studies in the Introduction¹

Study	Methodology	Limitation	Reported metrics for alite, belite & ferrite	Contribution from present work
Mastorakos et al. [14]	First principle modeling (1D dynamic CFD simulation)	 Lacks model validation 	NA	 Unprecedented accuracy validated on large-scale industrial data.
Hokfors et al. [21]	Physics-based modeling (process simulations using Aspen)	 Validated on just one data point from a plant Ignores influence of process parameters 	MAE_{alite} : 6.4 wt. % MAE_{belite} : 9 wt. % $MAE_{ferrite}$: 9.7 wt. %	 MAE_{alite}: 0.75 wt.% MAE_{belite}: 0.96 wt.% $MAE_{ferrite}$: 0.36 wt.%
Moses et al. [22]	Statistical modeling (quadratic regression)	 Small dataset (154 data points) Uses synthetic data; no plant validation Models only alite; excludes belite & ferrite 	R^2_{alite} : 1	 Considers impact of process parameters & kiln feed Uncertainty & extreme value testing
Ali et al. [30]	Machine Learning (purely data-driven neural network)	 Limited to predicting theoretical compounds (C_3S, C_2S, C_4AF), not actual clinker phases Lacks model interpretability Lacks uncertainty & rare event analysis No benchmarking across ML models 	NA	 Post hoc model explanation Plant-specific clinker equations Comprehensive ML benchmarking

References:

- Hub, C. S. Improving concrete sustainability through Alite and Belite reactivity. (2014).

2. Elakneswaran, Y. *et al.* Characteristics of ferrite-rich Portland cement: Comparison with ordinary Portland cement. *Front. Mater.* **6**, 97 (2019).
3. Zaki, M. *et al.* Cementron: Machine learning the alite and belite phases in cement clinker from optical images. *Constr. Build. Mater.* **397**, 132425 (2023).
4. Deja, J., Uliasz-Bochenczyk, A. & Mokrzycki, E. CO₂ emissions from Polish cement industry. *Int. J. Greenh. Gas Control* **4**, 583–588 (2010).
5. Davidovits, J. Global warming impact on the cement and aggregates industries. *World Resour. Rev.* **6**, 263–278 (1994).
6. Andrew, R. M. Global CO₂ emissions from cement production. *Earth Syst. Sci. Data* **10**, 195–217 (2018).
7. Li, C. *et al.* CO₂ emissions due to cement manufacture. in *Materials Science Forum* vol. 685 181–187 (Trans Tech Publ, 2011).